# Polymeric Microspheres Designed to Carry Crystalline Drugs at Their Surface or Inside Cavities and Dimples

**DOI:** 10.3390/pharmaceutics15082146

**Published:** 2023-08-15

**Authors:** Meitong Shen, Ling Zheng, Leo H. Koole

**Affiliations:** Innovative Bioengineering Laboratory for Ocular Drug Delivery, School of Ophthalmology and Optometry, Eye Hospital of Wenzhou Medical University, Wenzhou Medical University, 270 Xueyuan West Road, Wenzhou 325027, China; shenmeitong_eye@126.com (M.S.); zhengling180@126.com (L.Z.)

**Keywords:** drug delivery, microspheres, cavity, suspension polymerization, minimally invasive therapy, keratitis, transarterial chemoembolization

## Abstract

Injectable polymer microparticles with the ability to carry and release pharmacologically active agents are attracting more and more interest. This study is focused on the chemical synthesis, characterization, and preliminary exploration of the utility of a new type of injectable drug-releasing polymer microparticle. The particles feature a new combination of structural and physico-chemical properties: (i) their geometry deviates from the spherical in the sense that the particles have a cavity; (ii) the particles are porous and can therefore be loaded with crystalline drug formulations; drug crystals can reside at both the particle’s surfaces and inside cavities; (iii) the particles are relatively dense since the polymer network contains covalently bound iodine (approximately 10% by mass); this renders the drug-loaded particles traceable (localizable) by X-ray fluoroscopy. This study presents several examples. First, the particles were loaded with crystalline voriconazole, which is a potent antifungal drug used in ophthalmology to treat fungal keratitis (infection/inflammation of the cornea caused by penetrating fungus). Drug loading as high as 10% by mass (=mass of immobilized drug/(mass of the microparticle + mass of immobilized drug) × 100%) could be achieved. Slow local release of voriconazole from these particles was observed in vitro. These findings hold promise regarding new approaches to treat fungal keratitis. Moreover, this study can help to expand the scope of the transarterial chemoembolization (TACE) technique since it enables the use of higher drug loadings (thus enabling higher local drug concentration or extended therapy duration), as well as application of hydrophobic drugs that cannot be used in combination with existing TACE embolic particles.

## 1. Introduction

An emerging trend in the field of controlled local drug delivery appears to be the application of a pure crystalline drug as the release depot, rather than an amorphous drug that is embedded in, or attached to, a matrix of a (biodegradable) polymer [1,2,3,4,5,6,7]. A classic example is the use of the ophthalmic drug triamcinolone in the treatment of a broad spectrum of diseases of the retina [8]. Triamcinolone is a crystalline synthetic corticosteroid, and this offers the possibility to inject suspended particles that consist exclusively of the drug. This provides the basis for the commercial products Kenalog-40 and Triesence [8,9,10]. The use of crystalline triamcinolone brings the advantage that no encapsulating materials or expedients are required, i.e., there are no issues regarding remnants of polymer carriers post-delivery. Moreover, space is exploited to its full maximum, which is especially crucial in ocular drug delivery. Another example is the recent work of Farah and Domb, who studied the utility of crystalline paclitaxel and rapamycin at the surface of drug-eluting endovascular stents [7,11]. In the case of rapamycin, a rapidly dissolving polysaccharide coating was used to encapsulate and protect the drug prior to stent implantation. For paclitaxel, a carrier-free crystalline drug-coated stent was developed and described. Use of polymer-free stent coatings may be the answer to serious safety concerns about drug-coated stents in which a polymeric matrix is used. Polymers at the stent surface have been associated with elevated risk for late-stage thrombosis and inflammation, as well as with cracking, flaking, and delamination. More recently, Farah et al. also reported on the use of crystalline GW2580, a colony-stimulating factor 1 receptor inhibitor, to suppress the immune-mediated foreign body response to medical devices during long time intervals [2]. In non-human primates and in rodents, the foreign body response was substantially delayed (for 6 and 15 months, respectively). Taken together, the data on dexamethasone dimer [1], GW2580 [2], paclitaxel [3], rapamycin [7] and triamcinolone [8] point out that crystalline drugs combine a high drug density with slow topical delivery through dissolution. Dissolution appears to occur at the surface of the crystals and thus obeys zero-order kinetics, i.e., initial-burst release (which is often encountered with bio-absorbable polymer coatings) can thus be avoided.

These examples prompted us to investigate whether crystalline drug formulations could also be used advantageously in transarterial chemoembolization [11,12]. In TACE, drug-eluting embolic (DEE) microspheres/beads serve to locally stop arterial blood supply to tumors and as a drug delivery vehicle. DEE microparticles increase drug concentrations in tumors and reduce systemic drug exposure compared to conventional drug delivery methods. This strategy has proven particularly suitable for transarterial chemoembolization of hepatic malignancies [13,14,15,16]. Currently, there are seven commercial products for TACE with DEE microparticles [17,18,19,20,21,22,23]. While these products differ considerably regarding the polymer biomaterials (i.e., cross-linked polyvinyl alcohol, poly-methylmethacrylate coated with poly(bis-[trifluoro-ethoxy]phosphazene), acrylic acid (sodium salt)-vinyl alcohol copolymer, and poly(ethylene glycol)/3-sulfopropyl acrylate), they have many features in common. Notably, (i) all are loaded with the drug at the point through mixing of the particles with a drug solution, and (ii) loading of the drug occurs by an electrostatic (ionic) attraction between positively charged drug molecules and negatively charged sulfonate or carboxylate groups in the polymer structure. It follows that loading and unloading of such DEE microparticles is highly dependent on the medium, especially on the presence/absence of electrolytes, osmolarity, and pH. Release of the drug mostly passes an initial-burst phase, as is evident from release studies in vitro and in vivo [18,24].

To investigate the possibility of loading embolic particles with crystalline drugs we decided to use microspheres, which were also used in our previous work on TACE [25,26,27,28]. The biomaterial was a poly(methacrylate)-type three-dimensional network that was prepared from four different reactive monomers, i.e., methylmethacrylate, 2-hydroxyethylmethacrylate, tetraethylene-glycol dimethacrylate (the crosslinker), and 4-(iodo-benzoyl)-oxo-ethylmethacrylate (the building block that renders the material radiopaque). The biomaterial provided the basis for the bland-embolization product X-Spheres, which received the CE mark (in Europe) in 2012, i.e., the biomaterial passed extensive testing for its biostability and biocompatibility in the past. Recently we reported on non-spherical particles, which consisted of the same biomaterial and contained a relatively large cavity [28]. While we could show that the cavity of such particles could be partially filled with the crystalline drug (voriconazole was used as an example), we realized that round beads are preferred for TACE [29,30,31]. Hence, our specific aim was to study possible ways to obtain spherical DEE microparticles loaded with the crystalline drug, find ways to maximize the drug payload, and study the release of the drug in vitro.

Herein we first provide a powerful yet straightforward method to produce spherical DEE embolic particles having a crystalline drug attached to their surface. Binding of the crystals to the particle’s surface is through physical interactions; the crystals at the surface connect with other crystals that reside in the interior of the microparticles [28]. Thus, we effectively use the fact that the microparticle’s polymer network contains micropores. The drug crystals appear to be firmly attached. Second, we also provide a method to produce near-spherical microparticles that have a cavity or one or more dimples. To the best of our knowledge, particles of this size having one or more cavities or dimples have not been described previously. It was found that dimples and cavities can also be filled with drug crystals, at least partially. Then, some of the drug crystals are located at or near the surface, and some of the drug crystals reside in the dimple or cavity of such a microsphere. These particles are of interest, since they provide new possibilities to increase the particle’s payload, i.e., to extend the time interval during which the drug is locally available, or to apply hydrophobic drugs, which are currently difficult to combine with TACE.

We set out to study the drug-loaded spherical and near-spherical/cavitated particles with the drug voriconazole. We have a longstanding interest in this drug [29], and we are working on new technologies for sustained administration of voriconazole to the eye’s tear film, thereby using microsphere technologies. Furthermore, we considered this drug as an adequate model for many cytostatic agents that are used in combination with TACE, in view of its comparably poor solubility in aqueous media and its molecular mass. Two major advantages of using voriconazole are (i) its low toxicity (compared to most cytostatic agents), which allowed us to work with it easily and freely, and (ii) its composition, especially the fact that voriconazole contains fluorine. This facilitates analysis of the loaded particles (vide infra).

## 2. Materials and Methods

### 2.1. Materials

Methyl methacrylate (MMA), hydroxyethyl methacrylate (HEMA), ethylene glycol dimethacrylate (EGDMA), 4-iodobenzoyl chloride, triethylamine, di-isopropyl ether, poly(vinylalcohol) [(CH_2_-CHOH)_n_, 98–99% hydrolyzed], poly(vinylpyrrolidone) [(C_6_H_9_NO)_n_, BASF K-30], poly(ethylene glycol) [HO(CH_2_CH_2_O)_n_H; Mn = 1000] and tert-butyl 2-ethylhexaneperoxoate (initiator) were purchased from Macklin Biochemical Co., Ltd. (Shanghai, China). MMA was distilled at atmospheric pressure and stored at −20 °C prior to use; the other chemicals were used as received. 4-Iodobenzoyl-oxo-ethyl methacrylate (4-IEMA) was prepared as described previously [25,26]. It is a crystalline white powder with melting point 131 °C. The drug was purchased from Hangzhou Hyper Chemicals Ltd. (Hangzhou, China).

### 2.2. Microsphere Synthesis

All operations were carried out in a well-functioning fume hood and all normal safety precautions were taken. First, a stock of the detergent solution was made by dissolving poly(vinyl alcohol) (64.0 g), poly(vinylpyrrolidone) (29.0 g), and poly(ethylene glycol) (48.6 g) in 2000 mL of water. Mechanical stirring during 24 h at ambient temperature was required to completely dissolve all poly(vinyl alcohol). The stock was stored at 4 °C.

Detergent solution (400 mL) was transferred into a 1 L round bottom flask. The contents of the flask were magnetically stirred (500 rpm), and heated to 50 °C. The monomer mixture [consisting of MMA (4.00 g, 40.0 mmol), HEMA (0.52 g, 4.0 mmol), EGDMA (3.96 g, 20.0 mmol), 4-IEMA (3.60 g, 10.0 mmol), and initiator (0.43 g, 2.0 mmol)] was added dropwise (polypropylene pipette), the temperature was raised to 90 °C and maintained during 60 min. Then, heating was turned off, cold water (200 mL) was added, and the reaction mixture was allowed to cool to room temperature. Particles were worked up through decantation, repeated washing with water (>6 times), washed with ethanol 96% (once), and washed with water again (3×). Clean particles were collected on a Petri dish and dried overnight at 37 °C (oven).

In a subsequent series of experimental synthesis runs the protocol was changed in 2 respects: (i) an overhead mechanical stirrer was used (stirring speed: 250 rpm); (ii) the reaction was carried out at a higher temperature (95 °C). The remainder of the protocol stayed unchanged. The adapted protocol yielded mainly microparticles with dimples or cavities.

All dry particles were size-sorted on a sieving machine equipped with 6 sieves (apertures 800, 700, 500, 300, 200, and 100 micrometer). Particles on each sieve were collected and stored.

### 2.3. Microparticle Characterization

Light microscopy. Microscopic images were made with an Olympus CKX53 microscope (Tokyo, Japan) equipped with a BGIMAGING C20 camera/Sony Exmor CMOS sensor (Tokyo, Japan).

Scanning Electron Microscopy (SEM). Scanning electron microscopic (SEM) images were recorded using a Hitachi SU 8010 instrument (Tokyo, Japan) at an acceleration voltage of 5 kV and a working distance of 10 mm. Particles were mounted on a stainless-steel stub using double-sided tape and metalized with Pt using a Leica EM ACE600 sputter coater. The thickness of the Pt layer was approximately 5 nm (sputter time 200 s). Energy dispersive X-ray spectroscopy (EDX) experiments were run on the same instrument, now operating at 20 kV, with spot size 6 and a working distance of 10 mm.

Elemental analysis: The microparticles were analyzed for their content of C, O, and I. The results for the microparticles, which were synthesized according to route A were mass % C = 53.4; mass % O = 29.4; mass % I = 10.2. For the microparticles synthesized according to route B higher mass % values for iodine were found consistently. Typical results from 3 different synthesis routes B were (a) mass % C = 51.4; mass % O = 29.3; mass % I = 12.3; (b) mass % C = 51.1; mass % O = 28.5; mass % I = 14.3; (c) mass % C = 50.1; mass % O = 30.7; mass % I = 14.1.

UV-Vis Spectrophotometry. Samples were analyzed with a Perkin-Elmer Lambda UV-Vis spectrophotometer in scan mode; absorption was recorded from 200 to 800 nm at room temperature. Quartz cuvettes with path length 10.0 mm were used. Drug cargoes were quantified via the reference curves of respective drugs in alcohol.

Attenuated Total Reflection-Fourier Transform Infrared (ATR-FTIR) Spectroscopy. Fourier transform infrared spectra were acquired in total reflectance mode on a Perkin-Elmer 2000-series instrument (Seer Green, UK). Spectra were collected at room temperature in absorbance mode in the wave number range of 600–1500 cm^−1^. The background was recorded under ambient conditions without the sample. For each spectrum, 16 scans were collected with a nominal resolution of 4 cm^−1^.

Micro-CT analysis. These experiments were performed at room temperature on a Bruker SKYSCAN 1276 micro-CT system (Bruker, Kontich, Belgium) using the following parameter set: X-ray tube voltage 65 kV, tube current 370 μA, pixel space size 8.5 μm, rotation step length 0.7 deg., scanning angle range 180 deg., P*_max_* 0.1. The total scanning time was approximately 35 min. Averaging over 2 acquisitions was applied. The sample consisted of approximately 25 selected particles; selection was conducted through light microscopy and careful handling with tweezers. We decided to use relatively large particles (diameter range 500–700 μm) for these experiments. Particles were mixed with a commercial cosmetic cream (1 mL, spatulum) and then transferred into an insulin syringe (inner diameter 4 mm). The plunger was used to push the suspension to the middle of the syringe. The cream caused only low attenuation of X-radiation (checked previously). The suspension of the particles proved to be stable under the experimental conditions. The SKYSCAN CTAn software package (Bruker, version 1.17, Singapore) was used (off-line) to reconstruct the projection images and to generate various different images (e.g., cross sections) for analysis.

X-ray Photoelectron Spectroscopy (XPS). XPS measurements were performed externally by the company HuaYou Testing & Technology Service Ltd., Guangzhou, China (www.cyu-china.com, accessed on 12 June 2023). The instrument was a Thermo Scientific K-alpha spectrometer, which is a fully integrated monochrome small-spot XPS system. Pass energy and analysis region were set at 100 eV and 400 × 400 μm, respectively. Further experimental parameters: lens = standard, number of scans = 1, total acquisition time = 68 s, energy step size = 1.000 eV, and number of energy steps = 1361. Sample temperature was in the range 23 ± 2 °C.

### 2.4. Loading of Particles with Voriconazole

A solution of the drug (250 mg) in DMSO (1.25 mL) was divided over two 1.5 mL Eppendorf centrifuge vials. Added particles (90–100 mg) readily sank to the bottom. Open vials were carefully transferred (tweezers) into a conical glass flask (50 mL, NS 19 neck), which was equipped with a three-way valve. One of the ports was connected to a vacuum pump. A vacuum was applied and air bubbles were seen to escape from the particles. The vacuum was released and applied several times until no more air bubbles evolved. Vials were removed from the conical flask, closed, and left to stand for 24 h. The supernatant was removed as far as possible (using a needle/syringe), and the particles were spread as much as possible inside their vial. Vials were placed horizontally in a refrigerator (4 °C) for 1 h. Then cold water (1 mL, 4 °C) was added, and the vials were then left to stand at room temperature for several hours. Vials were gently shaken and the particles were washed with water (>6 times), collected on a Petri dish and allowed to dry overnight (37 °C, oven). The dry particles were transferred to the sieve from which they were collected originally (i.e., particles 500–700 μm diameter range were returned to the 500 μm aperture sieve, and 300–500 μm diameter range were returned to the 300 μm aperture sieve). Short machine sieving (2 min), which effectively separated the loaded particles from unbound and loosely bound drug crystals, afforded the desired drug-loaded microspheres [28].

### 2.5. Antifungal Experiment

These experiments were performed with voriconazole-loaded particles (diameter range 300–500 μm); unloaded particles of the same size and from the same synthetic batch served as negative controls. Clinical isolates of Fusarium spp. or *Aspergillus* spp. were used, i.e., these fungi had been isolated from patients’ infected eyes and preserved for research purposes. The densities of the conidial or sporangiospore suspensions were read on a spectrophotometer (530 nm, 1.0 cm light path) and adjusted to an optical density in the range 0.09–0.13 for *Aspergillus* spp. and 0.15–0.17 for *Fusarium* spp. Fungi were first spread on the agar surface (inoculating loop, streak plate method) [32,33]. Four Petri dishes were prepared, 2 with *Fusarium* spp., and 2 with *Aspergillus* spp. Ten drug-loaded microspheres and 10 empty microspheres were carefully placed on the agar layer of each dish, forming 2 hexagons (6 particles) + 2 triangles (3 particles) + 2 single particles (vide infra). Then, the dishes were placed in an incubator (35 ± 0.5 °C, 60% relative humidity) and examined/photographed daily. Zones of inhibition around the drug-loaded rods could be observed clearly.

### 2.6. In Vitro Drug Release

In vitro drug release experiments were performed on four types/categories of voriconazole-loaded microparticles, i.e., small size (300–500 μm diameter range) and spherical (category 1), large-size (500–700 μm diameter range) and spherical (category 2), small size (300–500 μm diameter range) and cavitated (category 3), and large-size (500–700 μm diameter range) and cavitated (category 4). These were carried out at 37 °C, which was chosen to correspond to normal human body temperature, during 96 h. First, triplicate 10 mg measures of drug-loaded microparticles were suspended in 10 mL of PBS, using 15 mL screw-capped tubes. The sample tubes were then placed on an incubator/orbital shaker rotating at 80 rpm and maintained at 37 °C. After certain time intervals (15′, 30′, 45′, 1 h, 90′, 2, 4, 6, 8, 14, 19, 24, 48, 72, and 96 h), the shaker was stopped, and 1.0 mL of the supernatant was removed from each tube. Freshly prepared drug-free PBS (1.0 mL) was then used to replace the removed supernatant to conserve sink conditions in each tube. The test samples were swirled and placed back in the shaker incubator for the continuous release study. The amount of released drug in each of the supernatant samples was assessed using an HPLC technique, thereby using (i) a procedure from the literature [34] and (ii) a calibration curve to determine the concentrations of voriconazole released from their respective drug-loaded microparticles. All HPLC analyses were performed with an Agilent 1260 Infinity II HPLC system (Agilent Technologies, Waldbronn, Germany) equipped with ChemStation software (version 6), G7111A 1260 Quaternary Pump VL, G7114A 1260 VWD detector, G7129A 1260 vial sampler, and a Poroshell 120 EC- C18 (4.6 mm I.D. × 150 mm, 4 μm) reversed-phase column (Agilent Technologies, St. Louis, MI, USA). All samples were filtered through a 0.1 μm Biofil syringe filter membrane prior to injection.

## 3. Results and Discussion

### 3.1. Microsphere Synthesis and Observations Made by Scanning Electron Microscopy (SEM)

The microparticles of this work were synthesized through suspension polymerization according to two different protocols. First, magnetic stirring was applied, yielding mainly spherical particles (Figure 1a) consisting of a 3D cross-linked polymer network; route A in Figure 2. The microspheres were size-sorted through sieving. Note that our monomer cocktail did not contain any porogen. Rather, MMA served as a solvent for the other monomers (notably 4-IEMA), while being one of the reactants itself. Our formulation led to formation of smooth polymer microspheres with a porous outer surface; pores had varying sizes with the largest diameter below 0.5 μm (Figure 1b). The microspheres were also subjected to analysis of their cytotoxicity in vitro (CCK-8 method, direct contact, and Live-Dead assay). These experiments showed that the microparticles do not interfere with the growth of cells in their vicinity, see Appendix A. Note that these experiments were conducted with L929 (mouse fibroblast cells) as well as with human corneal epithelial cells. The latter are known to be highly susceptible to toxicity effects exerted by foreign materials. The data from elemental analysis revealed that the mass % of iodine in our particles was 10.2, which explained the X-ray visibility as well as the elevated density of the particles as compared to the polymer microspheres described in the literature. The calculated mass % of iodine in the monomer feed (i.e., the mixture of MMA, HEMA, TEGDMA, 4-IEMA and initiator) was 8.4, thus lower than 10.2, which pointed at partial evaporation of MMA during synthesis (see below).

Figure 1c shows the interior of a typical microsphere. We noted that the SEM data on our particles resembled microstructures as reported by Dubinsky et al. [35] for some of their skin-free poly(glycidyl methacrylate (GMA)-co-EGDMA) microparticles. Specifically, the internal structures of our particles and of poly(GMA-co-EGDMA) synthesized in the presence of diethylphthalate (DBP)/dioctylphthalate (DOP) 3:1 (*v*/*v*), 1:1 (*v*/*v*) or DOP alone (see Figure 1c–e in the paper by Dubinsky et al. 2009) were highly similar [35]. This analogy suggested that χ-induced syneresis is the mechanism of pore formation. Furthermore, the periphery of our particles bore close resemblance to the microporous surfaces of poly(GMA-co-EGDMA) prepared in the presence of DBP or DBP/DOP 3:1 (*v*/*v*), see Figure 2b,c in ref. [35].

We studied the polymerization process further by changing several reaction parameters and observing the effects on the particles generated. Thus, we discovered that running the reaction while stirring mechanically and at temperatures of 95 °C and higher led to the formation of microspheres with aberrant geometries, notably microspheres with a cavity, as shown in Figure 1d–g, route B in Figure 2. These peculiar microparticles caught our attention since they combined two desired features: (i) near-spherical geometry and (ii) the presence of a cavity which can—in principle—be used as a temporary depot for (sustained) local drug delivery.

We assume that formation microparticles with a cavity (Figure 1d–j shows various typical examples) were related to the fact that the reaction was run close to the boiling temperature of MMA (100 °C). In each droplet, the polymerization reaction gradually proceeded from the periphery to the core. Since the reaction was exothermic and was run at 95 °C, it is likely that the core of each droplet heated up >100°. Hence, the MMA at the center of the particles was forced to evaporate, thus generating internal pressure. We believe that this led to temporary expansion of the particle, local rupture of the elastic shell, and escape of MMA vapor. Note that the elemental analysis data also pointed at partial evaporation of MMA (see above); the particles as prepared by route A (Figure 2) had an iodine content of 10.2 mass % (see above), while the particles as prepared according to route B had a higher iodine content (approximately 14%). Venting of the internal pressure allowed for relaxation of the elastic shell back to its spherical geometry, while the cavity that was left after the escape of gaseous MMA could be filled by the influx of detergent. The polymerization ended slightly later, particularly through a reaction of the less reactive pending double bonds (the second methacrylate groups of the cross-linkers), i.e., the cross-link density increased substantially at the end of the reaction. Interestingly, we observed several other aberrant particle geometries in support of our reasoning. First, some flask-like structures as shown in Figure 1g were encountered. It seems to us that such a geometry supported the idea of gaseous escape from the interior of the particle, but now with incomplete relaxation of the structure. Second, we also observed several near-spherical cavitated particles as shown in Figure 1h–j in which a small microsphere sat inside the cavity. We assumed that such a small microsphere was a daughter microsphere, which was generated in situ after relaxation of the original (parent) particle and influx of detergent into the cavity as postulated above. Phase separation of intruded aqueous detergent solution and unreacted monomer—still in the cavity—formed a new suspended droplet and hence a descendant microsphere. To the best of our knowledge, marsupial particles of this kind have not been described hitherto.

We note that our preparative route to particles with a dimple or with a cavity does not have a high degree of reproducibility, i.e., the yield of these particles versus “normal” microspheres varied considerably between different synthesis runs. This may reflect the complex and chaotic nature of the agitated suspension polymerization process in which monomer droplets are not only generated, but also made to collide and split into smaller droplets [36]. Under our experimental conditions the situation is even more complex, since (according to the mechanism as described above) the monomer droplets also undergo partial evaporation and burst while being converted into cavitated or dimpled polymerized particles. Small changes in the experimental parameters (especially stirring parameters such as depth of the stirrer in the reaction flask) can therefore have a profound impact on the outcome.

We were unable to find comparable asymmetric microparticles in the literature, especially regarding particle size. Several papers described nanoparticles with a dimpled structure. For example, Kegel et al. prepared and studied 3D-cross-linked polystyrene particles with a dimple structure; these particles had a diameter in the range 0.5–2 μm, i.e., substantially smaller than ours [37]. Other examples were found in the work of Tong and Gao et al. [38], who reported on particles with a discocyte (biconcave) geometry that resembled red blood cells (RBCs); these particles had a diameter of approximately 6.7 μm, which was still one order of magnitude smaller than ours. Doshi and coworkers described biocompatible particles with an RBC geometry consisting of a cross-linked layer-by-layer structure of either hemoglobin and bovine serum albumin (BSA), or poly(allylamine) and BSA, and a size in the range 7 ± 2 μm; these particles also appeared promising as vehicles for controlled drug delivery [39]. Zhang and coworkers also reported on 3D-cross-linked poly(styrene) particles with an RBS-like geometry (and also dimpled structures), which could be prepared on a large scale [40]. In their case, the non-spherical geometries arose from asymmetric shrinkage of a growing cross-linked network during phase separation. Meester and Kraft also described cross-linked polystyrene colloids with different (and tunable) sizes of the dimples; these particles were also substantially smaller than the particles presently described in this study [41].

### 3.2. X-ray Photoelectron Spectroscopic (XPS) Analysis

We analyzed our voriconazole particles further with XPS. This enabled identification and quantification of all elements (except hydrogen) present at the surface (depth 1–10 nm). The full XPS spectrum of our pristine microspheres is shown in Figure 3a. The expected XPS peaks (I3d, O1s and C1s) were observed. Figure 3b,c is discussed below.

### 3.3. Micro X-ray Computed Tomography (Micro-CT)

Microparticles with a cavity were further examined with micro-CT. This was possible because of their radiopaque nature owing to the presence of iodine in the macromolecular structure (intrinsic radiopacity). Hence, micro-CT provided a unique opportunity to “look inside the particles” and study the geometry of the cavities. Figure 4 shows a set of representative images obtained through micro-CT. The most important added value of these experiments lay in the fact that information was obtained about the size and depth of the cavities; SEM could not provide this. Figure 4a–d shows representative CT images of four individual cavitated microspheres at a surprisingly high level of detail and resolution. The cavities show a marked variation regarding their size and depth. The bottom row shows one of the microspheres in 10 consecutive planes (slices) from the data set. In this case, the near-spherical cavity protrudes to approximately 1/3 of the diameter of the microsphere. Note that it is the radiopaque nature of the polymer material that enabled us to make such detailed observations.

### 3.4. Voriconazole-Loaded Particles

Spherical microspheres and microspheres containing a cavity were loaded with voriconazole (vide supra). After drying and mechanical challenge during subsequent sieving, the spherical particles carried needle-shaped crystals of the drug attached to their surface. The cavitated microspheres had the drug crystals both at their surface, and inside the cavities (Figure 4a–d). FTIR spectral data (see Appendix A) were consistent with the molecular structure but unfortunately could not provide evidence for the presence/absence of voriconazole crystals at the surface of the microparticles. Spherical particles loaded with voriconazole were studied further through XPS and EDX-SEM experiments. Figure 3b, c shows the XPS subspectra in the regions 697–677 eV (around the fluorine F1s peak) and 409–392 eV (around the nitrogen N1s peak), respectively. F1s and N1s peaks were found; these signals corroborated that voriconazole resided at the surface of the particles. Note that (i) the F1s and N1s signals were much weaker than the I3d, O1s, and C 1s lines in Figure 3a, corresponding to the cross-linked polymer; (ii) the XPS spectra of the pristine microspheres did not show any peaks at these positions. Figure 5e–h shows SEM-EDX images of a voriconazole-loaded particle with a cavity; these images clearly revealed that the drug resided both at the particle’s surface and inside the cavity. Figure 5h (merge of Figure 5f,g) illustrates clearly where the drug was located, i.e., predominantly inside the cavity. SEM-EDX also revealed the presence of voriconazole outside the cavities, i.e., attached to the surface of the microsphere.

### 3.5. Antifungal Effect of Spherical Voriconazole-Loaded Particles In Vitro

Evidence of the release of voriconazole from the loaded spherical particles in vitro was derived from the experiment shown in Figure 6. Figure 6a shows the Petri dish at the beginning of the experiment, when diluted fungus (*Aspergillus* spp. in this case) was just distributed (seeded) on the agar layer on the Petri dish, and when 20 particles were carefully deposited on the agar layer as indicated: 10 drug-loaded particles were deposited on the left half of the dish, in a group of 6 (hexagon) + a group of 3 (triangle) + 1 single particle (black arrow heads in Figure 6b). On the right half of the dish, 10 pristine (unloaded) particles were deposited likewise (white arrow heads). Figure 6b, displaying the dish after two days, shows that so-called zones of inhibition were formed around the drug-loaded particles. These zones revealed that fungus around the drug-loaded particles was killed. The diameter of the zones decreased as follows: zone around the hexagon of six loaded particles > zone around the triangle of three loaded particles > zone around the single particle (Figure 6b), revealing that the fungus was eradicated in a dose-dependent manner. The fungus did not respond to the presence of the unloaded particles (negative controls on the right half of the dish). Figure 6c is analogous to Figure 6b, with the only difference that the fungus was *Fusarium* spp.

Subsequently, we quantified the loading of the particles by means of UV spectrophotometry. A reference curve was constructed from voriconazole in ethanol samples of known concentration and dilution; absorption at 256 nm was measured. It followed that the loading of voriconazole was 70.3 ± 2.4 μg/mg of small microspheres (diameter 300–500 μm) and 79.2 ± 2.1 μg/mg of large microspheres (diameter 500–700 μm). Alternatively stated, the voriconazole-loading percentages for the small and large particles were 7.0 ± 0.2% and 7.9 ± 0.2%, respectively. These data are compiled in Figure 7 (categories 1 and 2).

### 3.6. Voriconazole-Loaded Spherical Particles with a Cavity, Quantification

Since the encountered 7–8% voriconazole loading of the spherical particles was rather modest, our hope was concentrated on the cavities and dimples; specifically on the question of whether these could serve to accommodate a larger drug cargo. The SEM images in Figure 5a–d demonstrate the presence of relatively large crystals inside cavities or dimples. We performed UV quantification of the loading of these particles, adopting the method as described above. This led to the outcome that the cargo-loading of the cavitated particles was only slightly higher. For the small cavitated particles (diameter range 300–500 μm) the loading was 92.6 ± 5.3 μg/mg of particles, whereas the loading was 107.1 ± 3.1 μg/mg for the large cavitated particles (diameter range 500–700 μm). In terms of percentages: voriconazole loading for the small and large cavitated particles was found to be 9.3 ± 0.5% and 10.7 ± 0.3%, respectively. These data are compiled in Figure 6 (categories 3 and 4).

### 3.7. Release of Voriconazole, Measured In Vitro

Figure 8 shows our data on drug release (in vitro) of voriconazole-loaded microparticles with and without cavities. These experiments were performed with 10 mg of the drug-loaded microparticles immersed in 2.0 mL of PBS at a temperature of 37 °C. Aliquots (50 microliters) of the supernatant were taken at regular time points, and concentrations of voriconazole were measured with HPLC. As seen in Figure 7 (especially in the insert), release during the first 2 h was, to a good approximation, linear with time i.e., the release process obeyed zero-order kinetics at this stage. Thereafter, between 2 and 48 h, release slowed down, and after 48 h, virtually no voriconazole was released. For our intended applications, we envisage that drug delivery during the first 1–2 h is most important.

For each of the four cases it is now possible to calculate the amount of voriconazole that was released in the plateau phase, as compared to the drug loading of the 10 mg of microparticles. For the small particles (300–500 μm) without cavities (yellow) it was found that 10 mg of loaded particles released approximately 530 μg of the drug, while the calculated drug loading was 703 μg; in other words: approximately 75% of the drug loading of the particles was released under these conditions. For the other three cases this percentage was: (i) large particles (500–700 μm) without cavities (red): 73%; (ii) small particles (300–500 μm) with cavities (green): 87%; (iii) large particles (500–700 μm) with cavities (blue): 83%. These data indicate that the cavitated particles actually offer two advantages: they not only accommodate more voriconazole, but they also release a larger proportion of the drug.

### 3.8. Drug Release Mechanism

Using crystalline drug formulations combined with embolic particles will alter the mechanism of drug delivery occurring from the arrested embolic particles inside the tumor. The release kinetics will now be determined by the nature of the drug’s crystal structure, rather than by ion exchange [42,43]. It is well known that crystal structures of hydrophobic compounds are usually compact and stabilized by interactions between hydrophobic parts of the molecule and intra- and intermolecular hydrogen bonding. Water molecules cannot easily penetrate such structures, i.e., dissolution generally happens at the periphery of such crystals; this mechanism is known as surface erosion. Release via dissolution from a crystalline structure involves three factors: (i) disruption of the crystal lattice (endothermic), (ii) disruption of hydrogen bonds in water (“cavitation”) to accommodate the solute (endothermic), and (iii) solvation of the drug molecule (exothermic). The combination of these factors determines the Gibbs free-energy change—and therefore the kinetics—of the dissolution/drug release process. Further engineering and optimization studies are required to better understand in situ release kinetics regarding, e.g., crystal dissolution and various pH conditions or the most relevant pH conditions. Comparisons with neat API and physical mixtures of API and excipients must be made as well.

## 4. Conclusions and Future Directions

This study disclosed a new practical method to load polymer embolic microspheres with a crystalline drug. It was demonstrated that this technology works best for compounds that have poor solubility in aqueous media. Furthermore, we reported the preparation and characterization of nearly spherical microparticles having a relatively large cavity or one or more dimples. While reports on nanoparticles and small microparticles with a dimpled structure can be found in the literature, we are unaware of cavitated or dimpled microparticles sized as big as the ones presently described. Hence, we believe that these structures are novel and original. Our concept is clearly dependent on the adherence of a crystalline drug to the microparticles, irrespective of whether this is occurring at the surface of the microspheres or inside dimples or cavities. In the case of voriconazole, which crystallizes as needles with a high aspect ratio, the crystals form connections (clear examples are seen in Figure 4a,d). We anticipate that such inter-crystalline connections cause the drug to remain inside the cavities. Likely, similar physical connections between crystals might also stabilize the binding of drug crystals to the surface of microspheres, especially if crystals that were formed inside the porous structure of the microsphere connect with drug crystals residing at the surface. These hypotheses merit further research. Our observation that the cavitated or dimpled microparticles can be endowed with crystalline drug delivering capacity leads to the expectation that such particles can help to enhance the dosing of chemotherapies in TACE. Likely, the new insights from this study open new options to combine TACE with anti-cancer agents that are only marginally soluble in aqueous media, provided that they are crystalline.

## Figures and Tables

**Figure 1 pharmaceutics-15-02146-f001:**
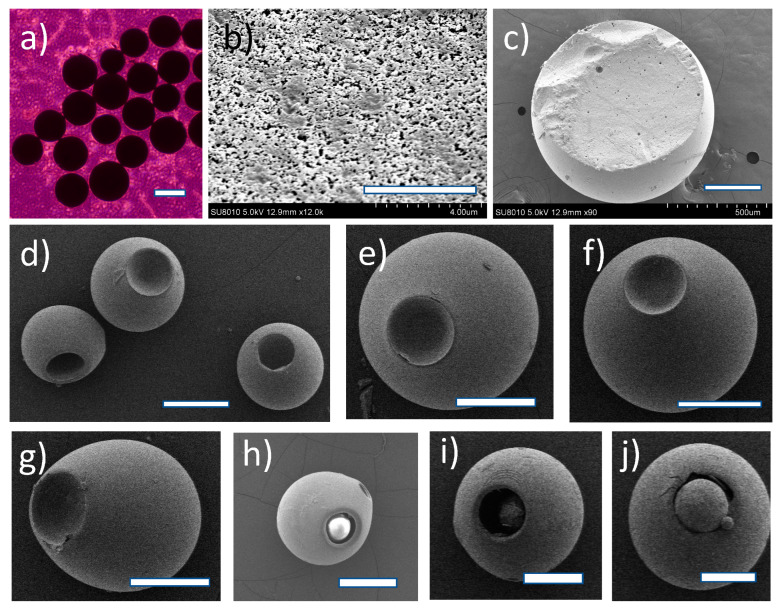
(**a**) Light microscopy of the microspheres obtained via the first synthesis method, see text. Scale bar = 300 μm. (**b**) SEM micrograph of the surface of microspheres as shown in (**a**). Scale bar = 4 μm. Pores in the surface are irregular with the largest diameter < 0.5 μm. (**c**) SEM image of the interior of a microsphere as shown in (**a**), cut with a razor knife. The large voids probably arise from oxygen formation during decomposition of the initiator (peroxide). Scale bar = 250 μm. (**d**) Representative examples of microspheres with a cavity obtained through the second synthesis protocol. Scale bar = 300 μm. (**e**,**f**) Examples of microspheres with a relatively small cavity (dimple). Scale bar = 300 μm. (**g**) Example of a microparticle with an elliptical flask-like geometry, also obtained through the second synthesis protocol. Scale bar = 250 μm. (**h**–**j**) Examples of microspheres with a relatively large cavity in which another microparticle (second generation, see text) is present. Scale bars are: 200 μm, 300 μm and 320 μm, respectively.

**Figure 2 pharmaceutics-15-02146-f002:**
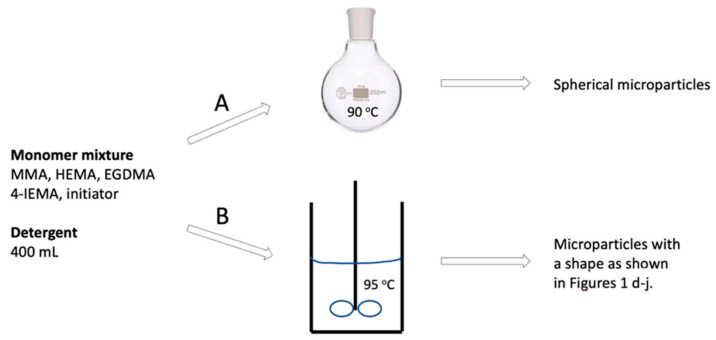
Schematic representation of the two synthesis routes used in this work to prepare 3D-cross-linked microparticles as microspheres (route A) or as dimpled/cavitated microspheres (route B).

**Figure 3 pharmaceutics-15-02146-f003:**
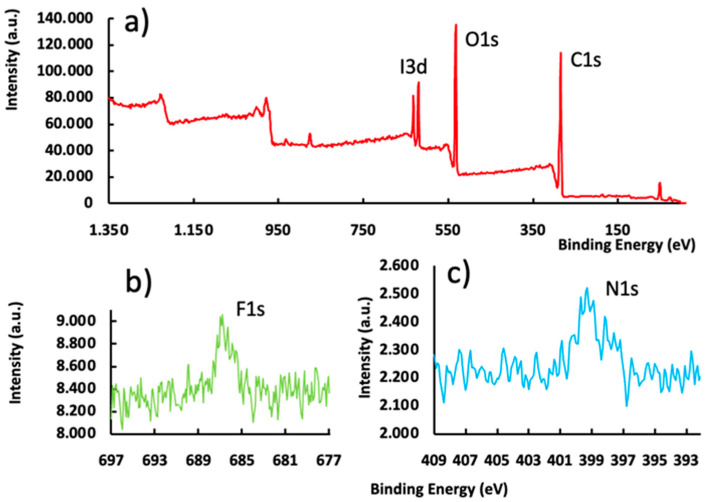
(**a**) Survey X-ray photoelectron spectrum (XPS spectrum) of microparticles as shown in Figure 1a. The spectral details confirm the identity and purity of the material. (**b**,**c**) F1s and N1s peaks as observed in the XPS spectrum of a microparticle as shown in Figure 1a, *after loading* with the drug voriconazole. The F1s and N1s peaks are not present in spectrum 2 (**a**) and can be ascribed to the presence of voriconazole (C_16_H_14_F_3_N_5_O) at the surface of the drug-loaded microspheres (vide infra).

**Figure 4 pharmaceutics-15-02146-f004:**
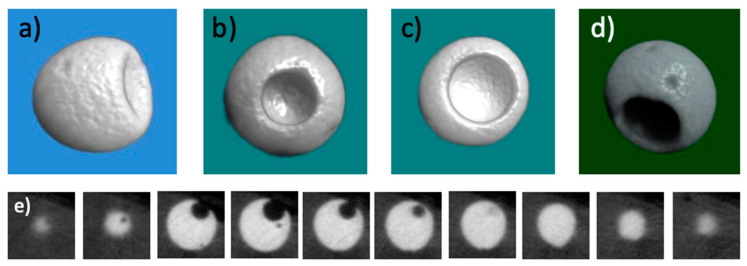
(**a**–**d**) Representative images of 4 different cavitated microspheres (not loaded with a drug) as generated by micro-computed X-ray tomography (micro-CT). These microspheres were selected from the largest regime of our synthesized particles; their diameter is approximately 600 μm. (**e**) Series of consecutive planar images of one of the microspheres; its diameter is also approximately 600 μm. Uniquely, this data gives insights into the depth and shape of the cavities.

**Figure 5 pharmaceutics-15-02146-f005:**
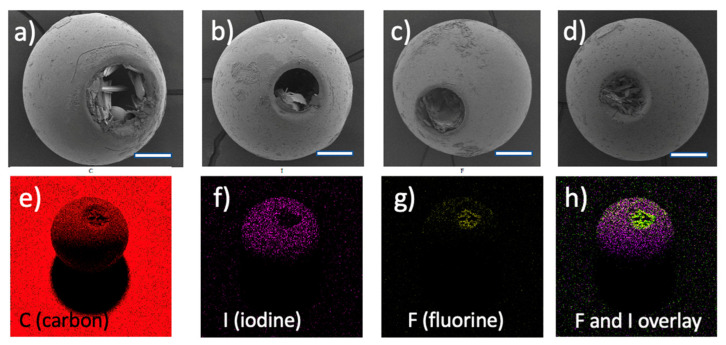
(**a**–**d**) Representative examples of microspheres with a cavity after loading with voriconazole. Crystals of the drug reside in the cavities, but also at the surface of the particles. Note that the size of the cavity varies considerably among the different particles. Scale bar = 100 μm in all 4 images. (**e**–**h**) Images from an EDX-SEM experiment on one of the voriconazole-loaded particles (diameter approximately 500 μm); (**e**): carbon, (**f**): iodine; (**g**): fluorine; (**h**) overlay of images (**f**,**g**). Note that iodine and fluorine are characteristic of the microsphere’s polymer and the drug, respectively. Image (**h**) reveals the presence of drug crystals primarily inside the cavity, but also on the particle’s surface.

**Figure 6 pharmaceutics-15-02146-f006:**
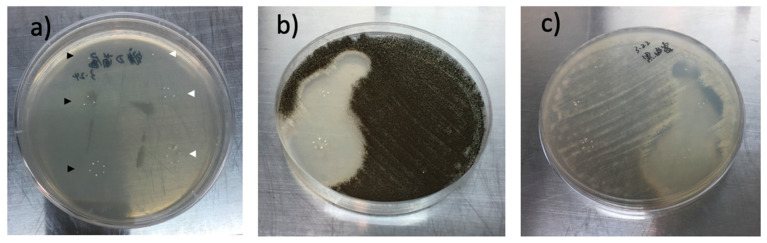
Microbiological experiments with the fungi *Aspergillus* spp. and *Fusarium* spp. (both clinical isolates) and release of voriconazole from the drug-loaded microspheres of this study. (**a**) Petri dish with an agar layer inoculated with the *Aspergillus* fungus. Ten voriconazole-loaded particles were carefully placed on the left half of the dish: 6 particles in a hexagon pattern, 3 in a triangle and 1 solitary; black arrow heads. On the right half of the dish, 10 unloaded particles were placed likewise (white arrow heads). The latter particles served as negative controls. (**b**) Petri dish of Figure 5a after 48 culturing. The fungus has grown (black area), except around the drug-loaded particles. Zones of inhibition with decreasing diameter in the series hexagon > triangle > solitary are seen, pointing at dose-dependent drug delivery via radial diffusion. The unloaded control particles (not visible here) have no effect on growth of the fungus. (**c**) As in (**b**), but now with the fungus *Fusarium* spp. which is optically transparent/whitish. The unloaded control particles (now visible) have no effect on fungus growth. In all cases the particle size was 300–500 μm (diameter range).

**Figure 7 pharmaceutics-15-02146-f007:**
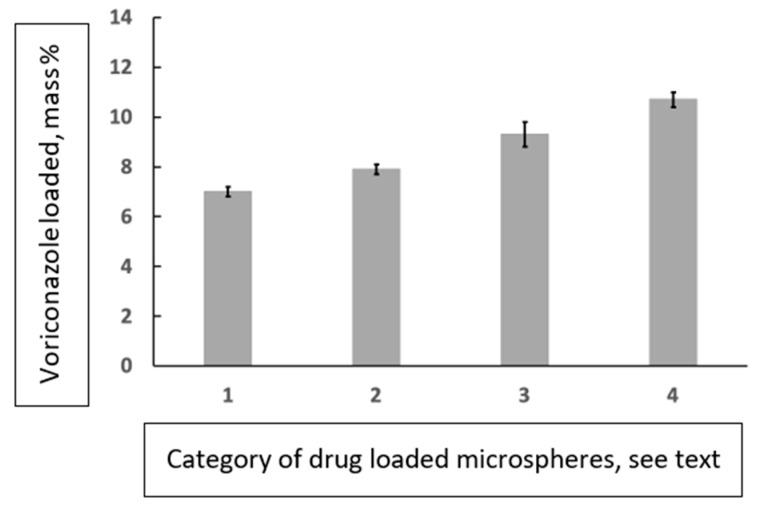
Histogram showing the drug loading (spectrophotometric determinations) of several microparticle/drug combinations studied; the vertical axis shows the drug-loading percentages, i.e., [drug loading (mg)/mass of the microspheres (mg)] × 100%. The categories (horizontal axis) are defined as follows: 1 = spherical microparticles within diameter range 300–500 μm loaded with voriconazole; 2 = spherical microparticles within diameter range 500–700 μm loaded with voriconazole; 3 = microparticles with a cavity within diameter range 300–500 μm loaded with voriconazole; 4 = microparticles with a cavity within diameter range 500–700 μm loaded with voriconazole.

**Figure 8 pharmaceutics-15-02146-f008:**
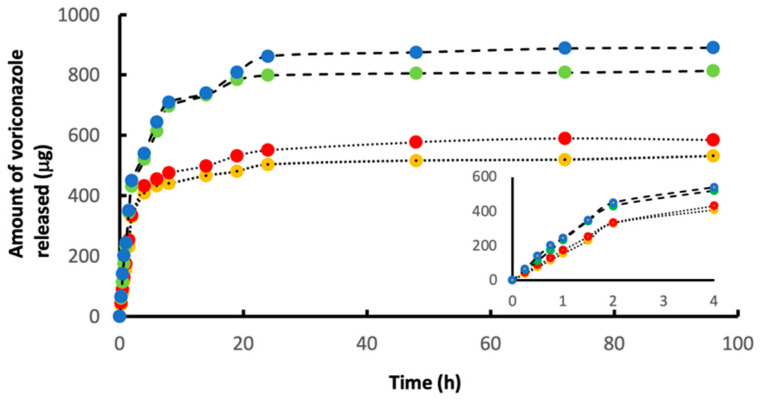
Data from in vitro experiments in which the release of voriconazole from the drug-loaded particles was studied (medium: PBS; temperature: 37 °C). The graphs plot the cumulative drug release versus time. The release curves show a typical initial rise and a plateau. The yellow and red data points refer to the spherical voriconazole-loaded small (300–500 mm) and large (500–700 mm) microspheres, respectively. The green and blue data points correspond to the cavitated voriconazole-loaded small (300–500 mm) and large (500–700 mm) microspheres, respectively. The data show the advantageous effect of the cavities: drug-loading capacity is enhanced by roughly 40%. Further analysis of the data revealed that the presence of the crystalline drug in the cavities also boosts the percentage of the drug that actually gets released (see text). The inserted graph in the lower right part of Figure 8 shows data on the release of voriconazole during the first 4 h in detail. To a good approximation, drug release occurs proportionally with time (zero-order kinetics) during the first 2 h.

## Data Availability

All data relevant to the publication are included.

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
