# Peer review of "Polymeric Microspheres Designed to Carry Crystalline Drugs at Their Surface or Inside Cavities and Dimples"

_pharmaceutics, 2023, doi:10.3390/pharmaceutics15082146_

Round 1
Reviewer 1 Report
see attached

Author Response
The point-by-point description of the changes we made is as follows:
Reviewer 1:
We agree with the reviewer that the clarity of the manuscript could be improved. As suggested by reviewer 1, we changed the reference system was changed, in alignment with the Pharmaceutics style.
The reviewer is right about the Supplementary Material. The Supplementary Material document (consisting of Part 1 and Part 2) is now included. Contents were carefully checked.
We followed the reviewer’s suggestion to include a Scheme which illustrates the 2 preparative routes as described in the manuscript. Now, the manuscript contains Scheme 1.
The reviewer is right that the different preparative pathways result in particles that have slightly different elemental compositions. We had these data, and this information is now included in the manuscript (pages 9, 13 and 15).
Regarding placement of the paragraph “Antifungal experiment”: the reviewers is right, and this was changed now. The “Antifungal Experiment” description is now 2.5, and hence comes after 2.4.
The reviewer is right about the ambiguity between voriconazole on one hand, and doxorubicin & irinotecan on the other hand. Hence, we followed their suggestion to remove the part of the manuscript that deal with doxorubicin & irinotecan. Voriconazole was loaded onto spherical particles as well as onto particles with cavity or dimples.
We agree with the reviewer that the data on particles loaded with doxorubicin & irinotecan were too preliminary. We followed the suggestion to delete these data and descriptions from the text.
In summary: we believe that we could respond to all the comments that were made by reviewer 1, except for the comment that more experimental research is required to establish the influence of several process parameters on the outcome of route B in Scheme 1. We agree with this point, and we did some work into this direction. We will continue, but in the limited time frame of this review we could not acquire conclusive data.
We believe that we could adequately address all the comments and suggestions made by the reviewers. We are most interested in their further remarks and suggestions.
We look forward to your reply and decisions.
On behalf of all authors,
Leo H. Koole, PhD
Professor of Biomedical Engineering
Innovative Bioengineering Laboratory for Ocular Drug Delivery
Eye Hospital of Wenzhou Medical University

Reviewer 2 Report
The topic taken up, which is controlled release of drugs, despite the fact that it has been developed for many years still presents new opportunities and challenges for scientists. The attempt to take an interest in not perfectly spherical objects made in this work seems to be very interesting.
The paper should be supplemented with the following:
1. One of the primary objectives of such research is to find ways to maximize the drug payload. Have there been studies comparing this for solid microspheres and the cavity microspheres proposed in this work? What were the results?
2. Is it possible and how to control the geometry and size of cavities in the mircosphere?
3. What is the effect of particle rotation (change in cavity position) on drug diffusion?
Author Response
The point-by-point description of the changes we made is as follows:
Reviewer 2.
This reviewer raised 3 questions, which we wish to answer as follows:
Ad 1: Yes, we have compared the drug payloads of the spherical particles, and the nearly-spherical particles with a cavity or dimple. The results are compiled in Figure 6. For the diameter range 300-500 micrometer, it is found that the drug-payload of the spherical particles is approximately 7.0 %, while the particles with a cavity carry approximately 9.3 %. This is an increase of 33 %. For the larger diameter range (500-700 micrometer), this increase was found to be 49 %. In other words, the particles with a cavity carry substantially more of the drug. This is now mentioned in the text.
Ad 2: It proved difficult to control the geometry and the size of the microparticles which are obtained via route B in Scheme 1. Reviewer 1 raised essentially the same point. The reason is that the suspension polymerization reaction, when run close to the boiling point of one of the reactants (MMA in our case) gets extremely chaotic. We are still working on this issue.
Ad 3: The effect of particle rotation/orientation on drug diffusion is difficult to characterize, and we have not yet done any experiments into this direction. For TACE techniques, there is no possibility to control the rotation/orientation of the particles in situ, but this might be technically possible in certain ophthalmologic operations. This must be explored further.
We believe that we could adequately address all the comments and suggestions made by the reviewers. We are most interested in their further remarks and suggestions.
We look forward to your reply and decisions.
On behalf of all authors,
Leo H. Koole, PhD
Professor of Biomedical Engineering
Innovative Bioengineering Laboratory for Ocular Drug Delivery
Eye Hospital of Wenzhou Medical University

Reviewer 3 Report
The manuscript presented for review presents research on the use of specially prepared microspheres containing dimples and cavities to bind with the crystallizing drug. The created release system is then evaluated to confirm the possibility of the controlled release effect of the selected drug (doxorubicin, irinotecan, voriconazole).
The work is competent, and contains; the exact procedures used for the synthesis and formation of microspheres, and their loading with selected drugs. The results of many studies using various instrumental techniques confirming the structure of the microspheres and the presence of the loaded drug have been presented. Requires only minor corrections and additions;
- introduce a little more description of the drugs selected for the test, why were they chosen as model drugs in the research? Description of the kinetics of crystallization of selected drugs, as well as the morphology of the forming crystals, differences and similarities. How can crystal morphology affect the process of bonding with the microsphere?
- Why were MMA/HEMA/EGDMA and PVA/PVP copolymer microspheres chosen as the carrier? Why were cross-linked polymers (hydrogels) chosen, which are difficult to biodegrade?
- There are no, even very simplified, in vitro studies of drug release kinetics from the described drug-loaded microspheres.
I did not notice any glaring spelling, syntax or other errors.
Author Response
The point-by-point description of the changes we made is as follows:
Reviewer 3.
This reviewer writes positive words about our manuscript, for which we are very grateful. Only minor corrections and additions are requested.
Since we deleted the parts on doxorubicin and irinotecan from the manuscript (as suggested by reviewer 1), we now only focus on the choice of the drug voriconazole in our reply to reviewer 3. Voriconazole was chosen since we have experience with this drug (see reference 29). It could be used as an optimal probe to compare spherical and cavitated microparticles regarding their drug-carrying features and capacities.
The choice of the particle’s formulation was also based on our experience. We know how to prepare these particles. The particles are stable, and they consistently feature a high level of biocompatibility (cell-friendliness) even against very sensitive cells such as human corneal epithelial cells, see Supplementary Data, part 2. In the past, the particles also have shown excellent biocompatibility in vivo, as is described in the text.
The reviewer is right that there are no in vitro drug release studies reported yet. This work is in progress in our laboratory.
We believe that we could adequately address all the comments and suggestions made by the reviewers. We are most interested in their further remarks and suggestions.
We look forward to your reply and decisions.
On behalf of all authors,
Leo H. Koole, PhD
Professor of Biomedical Engineering
Innovative Bioengineering Laboratory for Ocular Drug Delivery
Eye Hospital of Wenzhou Medical University

Round 2
Reviewer 1 Report
Authors have addressed most of the points that were remarked in my previous review.
Note: The Supplementary Material is still missing, or I have not been able to find it in the new submission, but I suppose that it will be available for the readers once the article is published.
Author Response
On behalf of the authors, I would like to thank Reviewer 1 for their comments. We will make sure that the Supplementary Material gets uploaded correctly.
Reviewer 3 Report
Unfortunately, I did not receive virtually any answer or correction in the text of the manuscript regarding my previous questions. So I have to stand by my earlier decision.
Author Response
Thank you very much for your message of July 6, in which you informed us about the feedback of 3 reviewers on the revised version of our manuscript “Polymeric microspheres designed to carry crystalline drugs at their surface or inside cavities and dimples” (Manuscript ID = 2441630), that we submitted on May 24 for publication in Pharmaceutics (Special Issue on Controlled Crystallization of Active Pharmaceutical Ingredients, Volume II).
We were pleased to see that reviewers 1 and 2 were -apparently- satisfied with the revisions we made.
This was not the case for reviewer 3 who stated that the revised version did not address the points he/or she had raised on the first original version of the manuscript. We acknowledge this vision of reviewer 3, and we apologize for this. In hindsight we realize that a few things went wrong on our side, as some of our intended replies did not end up in the revised version.
Hence, we have carefully examined the points that were raised by reviewer 3 again. We carried out additional experiments to evaluate in vitro release of voriconazole from our drug-loaded particles. We have addressed other issues that were put forward by reviewer 3, and the text was changed accordingly. New data were incorporated, as well as a new Figure (#7).
We believe that we could respond adequately to the points as raised by reviewer 3. We are grateful for the comments, and we believe that the quality of the paper has substantially improved because of this feedback.
The point-by-point reply to the comments is as follows:
- The reviewer asked for “a little more description of the drugs selected for the test, why were they chosen as model drugs in the research”. In the introduction section (page 3, marked green), we have added a sentence that further explains our choice for the drug voriconazole. We have a longstanding interest in this drug, and we are working on a microsphere-based technique for sustained delivery of it to the tear film. The reviewer may note that the revised version no longer contains data on microspheres loaded with cytostatic agents (irinotecan, doxorubicin). Data on such drug-loaded particles were deleted from the paper, due to comments from reviewers 1 and 2. Hence, the manuscript deals with 1 drug only (voriconazole), and we did not address the differences in crystal structures.
- Reviewer 3 asked us to comment on the bonding of drug crystals with the polymer microspheres, which may have dimples/cavities or be spherical, but always have a microporous structure.We described our ideas on this point on page 16 (marked green), pointing out that we believe that association/crosslinking between crystals (needles in the case of voriconazole) is essential in this respect. We acknowledge that bonding of drug crystals to polymer surfaces merits further research.
- Reviewer 3 correctly remarked that our work did not contain any data on drug release. Now, we have done a first series of drug-release experiments in vitro, and the results have been included in the manuscript (including a new Figure 7). These new data, however simple and preliminary, revealed that the presence of dimples/cavities does not only enhance the particles’ capacity to carry crystalline drug, but also enhances the percentage of the drug payload that eventually gets released.
We are grateful for the comments of reviewer 3, this feedback has really reinforced the manuscript.
